# SLC38A10 (SNAT10) is Located in ER and Golgi Compartments and Has a Role in Regulating Nascent Protein Synthesis

**DOI:** 10.3390/ijms20246265

**Published:** 2019-12-12

**Authors:** Rekha Tripathi, Kimia Hosseini, Vasiliki Arapi, Robert Fredriksson, Sonchita Bagchi

**Affiliations:** Department of Pharmaceutical Biosciences, Molecular Neuropharmacology, Uppsala University, 75124 Uppsala, Sweden; rekha.tripathi@farmbio.uu.se (R.T.); kimia.hosseini@farmbio.uu.se (K.H.); valia_arapi@yahoo.gr (V.A.); sonchita.bagchi@farmbio.uu.se (S.B.)

**Keywords:** solute carrier (SLC) family, amino acid transporters, SLC38A10, SNAT10, protein synthesis, transceptors

## Abstract

The solute carrier (SLC) family-38 of transporters has eleven members known to transport amino acids, with glutamine being a common substrate for ten of them, with SLC38A9 being the exception. In this study, we examine the subcellular localization of SNAT10 in several independent immortalized cell lines and stem cell-derived neurons. Co-localization studies confirmed the SNAT10 was specifically localized to secretory organelles. SNAT10 is expressed in both excitatory and inhibitory neurons in the mouse brain, predominantly in the endoplasmic reticulum, and in the Golgi apparatus. Knock-down experiments of SNAT10, using *Slc38a10*-specific siRNA in PC12 cells reduced nascent protein synthesis by more than 40%, suggesting that SNAT10 might play a role in signaling pathways that regulate protein synthesis, and may act as a transceptor in a similar fashion to what has been shown previously for SLC38A2 (SNAT2) and SNAT9(SLC38A9).

## 1. Introduction

Solute carriers (SLCs) are the largest group of transporters, with 460 members [1] (Perland and Fredriksson, 2017), (Kandasamy et al. 2018) divided into 52 families [2]. SLCs are localized to the plasma membrane [3,4], or intercellular organelle membranes (i.e., vesicles [5], mitochondria [6], peroxisomes [7], and lysosomes [8,9]). Eleven members of the solute carrier family 38 (SLC38) proteins are encoded by genes *Slc38a1-11*. In humans, this family was long thought to contain six members [10], until subsequent five members were identified [11]. The family is termed SNAT or sodium-coupled neutral amino acid transporters, number 1–11. This family of transporters is conserved in all mammals investigated [12] and the general substrate profile for most of the SLC38s are relatively similar based on the uptake data from heterologous expression systems [8,13,14,15,16,17,18,19,20]. Histological and cyto-chemical analysis of several proteins from this family revealed specific localizations within the brain [4,14,15,21]. SNAT1 is expressed in glutamatergic and GABAergic neurons, as well as in dopaminergic neurons of the substantia nigra, and in cholinergic motor-neurons [10]. In contrast, SNAT2 expression is ubiquitous and is found in all tissues tested [20,22,23]. In the CNS, SNAT2 can be detected in both neurons and glia cells but is more abundant in neurons [24]. Human SNAT4 has been found in liver [17] and placenta [25], and the rat orthologue is also found in skeletal muscle [17]. SNAT3 is abundant in the brain, liver, kidney, the eye, skeletal muscle, and adipose tissue [23,26,27]. In brain, it is mostly localized to astrocytes and endothelial cells of the blood-brain barrier [27,28,29]. SNAT5 is detected in multiple brain regions, liver, kidney, lung, colon, small intestine, stomach, and spleen [16]. SNAT6 is abundantly expressed in several tissues [11], but in brain, it is exclusively expressed in excitatory neurons [4]. Additionally, SNAT1 and SNAT2 have been shown to have specific patterns of expression in the CNS by *in situ* hybridization and immunocytochemical studies [20,30,31].

In addition to their role in amino acid transport [32], amino acid transporters have also been shown to function as nutrient sensors with the ability to control protein synthesis depending on the nutrient availability [33,34]. SLC38A9 has recently been identified as one of the components of amino acid-sensing Ragulator-Rag complex on lysosomes that is responsible for the activation of the mechanistic target of rapamycin complex1 (mTORC1) in response to amino acids [8,19,35].

One of the SLC38 family members, SLC38A10, has previously been identified in mouse brain tissue sections and it mediates bidirectional transport of glutamine, glutamate, and aspartate [36]. However, its function is still unknown. Here we characterize this transporter further in order to explore its subcellular localization, as well as its function(s) in the cell. We show that SLC38A10 (SNAT10) is present in both excitatory and inhibitory neurons. This transporter is expressed intracellularly, unlike many of its family members. Additionally, we report a marked reduction in protein synthesis in cells where *Slc38a10* is knocked down using siRNA, suggesting a probable role for SNAT10 in protein synthesis. 

## 2. Results

### 2.1. SNAT10 Is Expressed in Both Excitatory and Inhibitory Neurons

Mouse sections stained with a custom SNAT10 antibody [36] and Alexa Fluor 488 (green) secondary antibody, showed that SNAT10 is expressed in most neurons (Figure 1A) because the general neuronal marker NeuN co-stains with SNAT10 to a very high degree. SNAT10 immunoreactivity did not overlap with MAP2 immunoreactivity, showing that SNAT10 is not expressed in neuronal extensions, but is found in the soma (Figure 1B). SNAT10 immunoreactivity is not found in the same cells as GFAP immunoreactivity (Figure 1C), revealing low or completely absent expression in glia cells. Primary mouse cortical cell cultures, where the VIAAT expressing inhibitory neurons are genetically marked with enhanced green fluorescent protein (eGFP), was used for the immunostaining of SNAT10 with the same primary antibody as in Figure 1A–C [36] and Alexa Fluor 594 as secondary antibody. Red signals for SNAT10 were detected (Figure 1D, yellow arrows as example of positive cells) as well as green eGFP signals for inhibitory cells (Figure 1D, middle panel with white arrow). SNAT10 was found to be expressed in inhibitory neurons, as well as other neurons (Figure 1C, three color overlay). This, together with data in Figure 1A, shows that SNAT10 is expressed in the majority of neurons both excitatory and inhibitory.

### 2.2. The Subcellular Localization of SNAT10

The SLC38A10 expressing cell lines PC12 and N25/2 were stained with anti-SNAT10 antibody [36] with Alexa Fluor 488 and Alexa Fluor 594, respectively, to visualize its intracellular localization (Figure 2A,B). Interestingly, micrographs showed SNAT10 immunoreactivity in the cytoplasm, with significantly higher signal localized at the edges of the nuclei (Figure 2A,B). Signals were often found more prominent on one side of the nuclei. Transfected PC12 and HEK293 cell lines confirm intracellular localization of SNAT10. The subcellular localization of SNAT10 in PC12 cells was unique for this family of transporters. A co-localization study was performed to explore this further. PC12 and neuronal stem cells were transfected with a construct containing SNAT10, translationally tagged to eGFP and FLAG. Additionally, co-stained within Golgi 58K (Golgi marker) fluorescent micrographs from both the cell lines confirmed that SNAT10 is localized at one side of the nuclei (Figure 2H,I) suggesting expression in or near the Golgi apparatus. Major portions of SNAT10 colocalized within the Golgi.

### 2.3. Proximity Ligation Assay Exhibits a High Level of Interaction between SNAT10 and Golgi Proteins

Proximity ligation assay (PLA) revealed positive interaction between SNAT10 and Golgi protein 58k (Figure 2E) on wildtype mouse brain tissue sections. The number of interactions between these were significantly higher than what was observed between SNAT10 and SNAP25 (Figure 2F), which is a protein expressed mainly at the synaptic membrane, and in glia cells, and the soma (mainly the plasma membrane) of neurons. There was minimal interactions observed between SNAT10 and synaptophysin (Figure 2G), mainly restricted to the synapses [37], and found in all cells. Hence, SNAT10 is not expressed at the synapses or to any larger extent in the plasma membrane. PLA detects interactions of two proteins if they are in close proximity, specifically 30–40 nm apart. PLA relies on antibodies but is less prone to false positives compared to regular immunocytochemistry, because binding of two antibodies is required for a signal [4]. This indicates that expression of SNAT10 is intracellular and in proximity of Golgi, in neuronal cells. 

### 2.4. Knockdown of SNAT10 Affects Protein Synthesis

*Slc38a10* specific siRNA was used to knockdown expression of SNAT10 in PC12 cells. After 48 h of siRNA transfection, rate of protein synthesis was measured using an OPP labelling kit which incorporated OPP (O-propargyl-puromycin, OPP) into the C-terminus of translating polypeptide chains, which is further visualized using fluorescent Click-iT chemistry [38]. After 48 h of *Slc38a10* siRNA knockdown, a significant (*p* = 0.001) reduction in the protein synthesis rate was measured in the *Slc38a10*-specific siRNA treated cells compared to scrambled siRNA.

## 3. Discussion

Several members of the SLC38 transporter family have been studied extensively: SLC38A1-SLC38A9. SLC38A10 and SLC38A11 are yet to be characterized in detail. Here we aim to elucidate the tissue distribution of SLC38A10 (SNAT10), determine its cellular localization, and downstream effects of *Slc38a10* silencing, in PC12 cells, on protein synthesis.

Using double immunocytochemistry (ICC) (Figure 1A–C) and PLA (Figure 2E–G) we show that SNAT10 immunoreactivity is found in both excitatory and inhibitory neurons, at intracellular membranes close to the nuclei. To investigate the subcellular localization of SNAT10 we performed double ICC, with the KDEL (ER marker) and Golgi 58k (Golgi Marker) on cells transfected with a plasmid containing human SLC38A10 tagged with eGFP and FLAG epitope. SNAT10 expression overlapped with both Golgi and ER markers. This expression could be analogous to what has been found in hepatocytes where SNAT2 was found to be stored in the trans-Golgi network, and recruited to the plasma membrane in response to insulin stimulation [22]. SNAT10 is also predicted to have an intracellular N terminal, and extracellular C terminal of 722 amino acid residues [36]. The C and N termini play important roles with respect to its subcellular expression, stability, and cellular signaling [39]. In yeast, some transporters act as transceptors and play crucial role in signaling pathways [39]. The observed SNAT10 expression suggestions a potential role in Golgi membrane function, which are supported by the protein synthesis rate measurements in this study (Figure 3), suggesting SNAT10 could act as a transceptor in mammalian cells, with its main function in the Golgi. Of the investigated SLCs, the SLC35 family is also known to function in the Golgi. The SLC35 protein family encode nucleotide sugar transporters and transport nucleotide sugar substrates which are used for glycosylation of proteins. SLC35 transporters are exchangers and transport non-sugar bound nucleotides out of the Golgi lumen where those substrates are present in excess, and exchange these for sugar bound nucleotides. Apart from this, no SLCs are known to have a functional role in the Golgi membrane and the role of a non-general amino acid transporter such as SNAT10 in the Golgi membrane has not been established. Amino acids are crucial for protein synthesis, but one would expect that a transporter with a role in this process should have a general transport profile, being able to translocate all or most of the biogenic amino acids. It has, however, already been shown that SNAT10 has a substrate profile restricted to specific amino acids [36], and hence would most likely not have this function, unless this function is shared by several transporters which together cover import of all biogenic amino acids into the Golgi lumen. One other possible function of SNAT10, as indicated by our protein synthesis measurements, could be as a signaling role in the Golgi membrane. This form of signaling may require SNAT10 to sustain protein building and modifications. When we knocked down *Slc38a10* using siRNA, this signal was disrupted. A signaling role for SNAT10 would be analogous to the recently discovered function of SLC38A9, which was expressed on lysosomes and signal amino acid availability through the mTORC1 pathway [8,19]. A possible signaling role for SNAT10 needs to be further investigated. One other possibility is that SNAT10 translocated from the Golgi to the plasma membrane, making the Golgi localized SNAT10 function as a reserve pool, which is shifted to the plasma membrane under certain conditions. It could be that under the conditions we are culturing the cells; the proportion of SNAT10 found in the plasma membrane is too small to detect. This small amount of SNAT10 could however still be functionally relevant, and be part of the explanation for the effect on protein translation.

In conclusion, we have shown that SNAT10 expression is found in most neurons of the mouse brain. It has a unique intracellular localization in several cell lines, strongly suggesting expression in the Golgi. Silencing the expression of SNAT10 in PC12 cells resulted in the reduction of nascent protein synthesis, suggesting that SNAT10 might function as a transceptor.

## 4. Materials and Methods

### 4.1. Ethical Statement

Animal care procedures were approved by the local ethical committee (Uppsala djurförsöksetiska nämnd) within the district court in Uppland county (permit C134/14) and followed the guidelines of European Communities Council Directive (86/609/EEC).

### 4.2. Cell Cultures

Four different cell lines, used in this study, are described as follows:The immortalized rat adrenal gland cell line PC12 Adh (ATCC, CRL-1721.1) was cultured in ATCC-formulated complete growth media F-12K (catalog no. 30-2004) supplemented with 12.5 mL fetal bovine serum (FBS), 75 mL horse serum, 5 mL penicillin-streptomycin (Pen-Strep) to a 500 mL F-12K media flask. All cells were incubated at 37 °C with 5% CO2. The cells were seeded on glass slides (coated with 10 μg/mL poly-L-lycine) for 40 h prior to immuno-staining. All chemicals/media were from Gibco, Stockholm, Sweden, except another company name is stated. Cell passage nine was used for experiments.The immortalized embryonic mouse hypothalamus cell line N25/2ATCC (mHypoE-N25/2, Cellutions Biosystems Inc., Burlington, Ontario, Canada) was cultured in Dulbecco’s modified eagle medium (DMEM (+) 4.5g/L *D*-Glucose, (+) *L*-glutamine, (+) pyruvate) supplemented with 50 mL fetal bovine serum (FBS), 5 mL penicillin-streptomycin (Pen-Strep). Cell passage eight was used for experiments.The immortalized human embryonic kidney cell line (HEK293)ATCC, was cultured in Dulbecco’s modified eagle medium (DMEM [+] 4.5g/L *D*-Glucose, (+) *L*-glutamine, (+) pyruvate), supplemented with 50 mL fetal bovine serum (FBS) and 5 mL penicillin-streptomycin (Pen-Strep). Cell passage six was used for experiments.Neuronal stem cells were differentiated from human embryonic stem cell H9 line (Wi cell, Madison, Wisconsin, USA) for a duration of seven days using 1:49 ratio of Gibco Neurobasal^®^ medium and Gibco^®^ Neural Induction Supplement. After the cells were differentiated they were harvested and expanded in neural expansion medium. We prepared 50 mL of the medium each time consisting of 24.5 mL of Neurobasal^®^ Medium and Advanced™ DMEM⁄F12 (Thermofisher, Uppsala, Sweden). The cells were seeded on geltrex and splitted 1:5 ratio every 4–5 days. After the second passage post-thaw, they were used for the experiments.

### 4.3. Construct and Transfection Protocols

The FLAG tag (DYKDDDDK) was inserted in frame after the first start codon of human SNAT10 and cloned into PCDNA3.1 (Thermo Fisher) and SLC38A10-eGFP plasmid, where eGFP was cloned in frame at the C-terminal of human SLC38A10, was purchased from Addgene (plasmid # 62020) [40]. Transfections of this plasmid into N25/2 and HEK293 cell lines were performed using MaTra A reagent (IBAfect, Göttingen, Germany) according to the manufacturer’s instructions and cells were processed 24 h after each transfection. For all transfection experiments, the concentrations of the plasmid in serum free DMEM media was 1.2 ug/mL (0.6 µg per well).

For neuronal stem cell and PC12 cells transfection Lipofectmine 3000 was used according to the manufacturer’s instructions and cells were processed 48 h after each transfection. The concentrations of the plasmid was 1.0 ug/mL.(0.5 µg per well).

### 4.4. Mouse Primary Cortical Culture

C57Bl6/J female mice (Taconic M&B, Copenhagen, Denmark) were mated with VIAAT-eGFP heterozygous males and at embryonic day 15–16, the females were sacrificed by severing the spinal cord. Embryos were removed from the uterus and kept in cold HBSS buffer (Gibco, Stockholm, Sweden) during separation from the yolk sac and placenta. Embryos were decapitated before cortex dissection was performed in 1x phosphate-buffered saline (PBS) with 10 mM glucose, under a Leica CLS 100 LED microscope. The cortices were chemically dissociated in 10 µg/mL DNase (Invitrogen, Stockholm; Sweden) and 0.5 mg/mL Papain (Sigma-Aldrich, Stockholm; Sweden), diluted in PBS with 10 mM glucose for 30 min at 37 °C in presence of 5% CO_2_. Tissues were then rinsed in plating media DMEM-F12 containing 10% FBS, 2 mM *L*-glutamine, 1 mM Na-pyruvate, and 1% penicillin/streptomycin (all chemicals where purchased from Gibco/Invitrogen, Stockholm; Sweden). Afterwards they were mechanically dissociated by pipetting up and down with a glass Pasteur pipette and filtered through a 70 μm nylon cell strainer (BD, Stockholm; Sweden) to remove the remaining cell clusters. Finally, the cells were plated at a density of 7.5 × 10^4^ cells on Poly-*L*-lysine (Sigma-Aldrich, Stockholm; Sweden) coated cover slips (12 mm, #1.5, VWR, Stockholm, Sweden) and incubated for 3 h at 37 °C in presence of 5% CO_2_. Plating media was then replaced with growth media Neurobasal-A (Gibco, Stockholm; Sweden) with 2 mM *L*-glutamine, 1 mM Na-pyruvate, 1% penicillin/streptomycin, and 2% B27 (Invitrogen, Stockholm; Sweden). Two-third of the growth media was changed every third day and on the tenth day cells were rinsed with 37 °C PBS with 10 mM glucose and fixated in 4% PFA (Histolab, Sweden) for 10 min, followed by additional washes in PBS. Cells were kept in PBS until used.

### 4.5. Fluorescent Immunocytochemistry

Cells were rinsed with PBS and fixed in 4% paraformaldehyde (PFA, Sigma-Aldrich, St.Louis, Missouri, USA) for 15 min. Then the slides were pre-blocked with supermix (Tris-buffered saline, 0.25% gelatin, 0.5% Triton X-100) for 1 h at room temperature. Then the primary antibodies diluted in supermix were added to respective slides for overnight incubation at 4 °C. After repeated washings with 1X PBS (Sodium perborate), they were incubated with secondary antibody (always 1:400 dilutions in supermix when Alexa Fluor antibodies were used, see Table 1 for details) for 1 h at room temperature. After several washing steps, DAPI (1:3000 in PBS) was added for 10 min at room temperature. After washing, the slides were mounted in DTG media (with antifade (diazabicyclo (2.2.2) octane in 80% glycerol and 50 mM Tris pH 8.6) and photographed using a Zeiss AxioPlan 2 fluorescence microscope, connected to an AxioCamHRm camera. The micrographs were finally analyzed with Carl Zeiss AxioVision version 4.8 software (Zeiss, Oberkocken, Germany).

### 4.6. Co-Localization Immunostaining

After transfection of GFP- and FLAG-tagged SLC38A10 vector fixed cells were stained with ER and Golgi marker. In brief SLC38A10 eGFP/FLAG tagged vectors were transfected. Cells were fixed with 4% PFA and immunocytochemistry was performed as described in [41] with antibodies Golgi 58k (Abcam, Cambridge, United Kingdom) as Golgi marker and KDEL (Abcam) as ER marker. For localization of the transporters, images were acquired at the SciLifeLab BioVis Facility using confocal Zeiss ELYRA S.1 and the Zen black software (Zeiss, Oberkochen, Germany) and images were handled using ImageJ (Fiji edition, https://imagej.net/Fiji/). For fluorescent colocalization, images were taken using on an Olympus microscope BX53 with an Olympus DP73 camera and CellSens Dimension software.

### 4.7. Image Analysis for Immunocytochemistry

All other samples from immunocytochemistry experiments were photographed using a Zeiss AxioPlan 2 fluorescence microscope (20× and 40× magnifications on single plane) connected to an AxioCamHRm camera and the micrographs were analyzed with Carl Zeiss AxioVision version 4.8 software.

### 4.8. Proximity Ligation Assay (PLA)

The Duolink II fluorescence kit (orange detection reagents, Olink Biosciences, Uppsala, Sweden) was used to perform *in situ* PLA on fixed cells and/or paraffin embedded sections (as described before) according to manufacturer’s instructions [40,41,42]. The samples were blocked with blocking solution included in the kit for 30 min at 37 °C in a pre-heated humidity chamber. Then specific primary antibodies (see Table 1 for concentrations) diluted in antibody diluent included in the kit were added to the samples and incubated overnight at 4 °C in a humid chamber. After that PLA probes (PLUS and MINUS) were added for 1 h at 37 °C in a pre-heated humidity chamber. In our experiments, protein interactions were detected with combinations of anti-rabbit PLUS and anti-mouse MINUS or anti-mouse PLUS and anti-goat MINUS PLA probes. Then the detection protocol, including ligation and amplification, was followed. The ligation step was performed for 30 min at 37 °C and the amplification step was performed for 100 min at 37 °C both in a pre-heated humidity chamber. The slides were washed according to the manufacturer’s instructions with buffers provided in the kit, dried in the dark for 20 min, and mounted using a minimal volume of Duolink *in situ* mounting medium, containing DAPI. The edges were sealed using transparent nail polish, and were visualized using Zeiss Axioplan2 fluorescent microscope connected to an AxioCamHRm camera. All the micrographs were obtained by using Z-stack imaging function. A negative control was included without primary antibodies. The images were further analyzed and quantified using Duolink ImageTool (Olink Biosciences, Uppsala, Sweden) software.

### 4.9. Image Analysis for PLA

Z-stack images from different fields of each slide were taken using a Zeiss AxioPlan 2 fluorescence microscope (20× and 40× magnifications on single plane) connected to an AxioCamHRm camera and the micrographs were analyzed with Carl Zeiss AxioVision version 4.8 software.

### 4.10. Flouorescent Immunohistology

Fluorescent immunohistochemistry was performed according to [15], on paraffin-embedded mouse brain sections sectioned at 10 µm and stained with the custom made SNAT10 antibody detected with anti-rabbit Alexa 594 (red). Co-staining was with anti- NeuN (1:200; Millipore, Sweden), anti-MAP2 (1:250, Sigma, Stockholm, Sweden), and Anti-GFAP (1:200, Millipore, Sweden) and detected with secondary antibody conjugated to Alexa 488 (green). Sections were imaged with a Zeiss LSM700 confocal microscope.

### 4.11. Knock down of Slc38a10 in PC12 Cells

The PC12 cells were seeded onto 24-well plates, with up to 80–90% confluency. Then, 3 μL Lipofectamine^®^ RNAiMAX (Thermofisher, Waltham, Massachusetts, United States) was diluted in 50 μL Opti-MEM^®^ Medium (Life Technologies) and 10 pmol Slc38a10 siRNA (Ambion^®^, Life Technologies) was diluted in 50 μL Opti-MEM^®^ Medium separately. The diluted siRNA, for a final concentration of 0.01 pmol/mL in the culture, was then mixed with the diluted Lipofectamine^®^ RNAiMAX Reagent in a 1:1 ratio and was incubated for 5 min at room temperature. From the siRNA-lipid complex, 50 μL was added to the cells which were incubated for 48 h at 37 °C with 5% CO_2_. Transfection was performed with the same amount of control siRNA (negative control Silencer) (Ambion^®^, Life Technologies) as was used for SNAT10, with the same amount of lipofectamine and RNAiMAX as with SLC38a10 siRNA. Some wells were used as non-siRNA control, treated with transfection reagent only. The experiment was repeated twice and the data were normalized for cell number differences between untreated control cells seeded on the same set of plates. Verification of the knockdown RNA was done using Rneasy Mini kit (QIAGEN, Hilden, Germany) and converted to cDNA using high-capacity RNA-to-cDNA Kit (Thermofisher, Waltham, Massachusetts, United states). cDNA of *Slc38a10*, negative silencer and wild type were used in q-RT-PCR analysis with *Slc38a10* specific primer (Invitrogen, Uppsala, Sweden) and normalized with β-tubulin expression (Invitrogen).

### 4.12. Measurement of Protein Synthesis Using OPP Kit

PC12 Cells were plated on 24-well plate and transfected by RNAiMAX. After two days of incubation, cells were ready for nascent protein synthesis using Click-iT^®^ OPP 488 Kit (Invitrogen, Carlsbad, California, United States). On the day of experiment, component A was diluted 1:1000 ratio in complete growth medium in order to prepare 20 μM working stock solution. Cell medium was removed and 1 mL of working solution was added per coverslip. Twenty μM cyclohexamide solutions were also prepared as a negative control and were added to 1 cover slip per group of transfection. Cells were incubated for 30 min in the incubator at 37 °C. Cells were then fixed for 20 min using 4% PFA at room temperature. For cells to be permeable, 0.5% Triton^®^ X-100 (Sigma, Missouri, USA) in PBS (Thermofisher, Waltham, Massachusetts, United States) was added for 15 min. 1× Click-iT^®^ OPP reaction buffer additive was prepared by diluting 1:10 ratio of 10× solution in deionized water. In the next step, Click-iT^®^ Plus OPP reaction cocktail was prepared by adding, 1× Click-iT^®^ OPP reaction buffer (880 μL), copper protectant (component D) (20 μL), Alexa Fluor^®^ picolyl azide (component B) (2.5 μL) and Click-iT^®^ reaction buffer additive 1× (100 μL) (quantities are for 1 cover slip). Permeabilization solution was removed and cells were washed twice with PBS. Cells were incubated with 1 mL of the reaction cocktail per coverslip for 30 min at room temperature. Cells were washed twice with PBS and mounted using ProLong Gold antifade reagent with DAPI (Thermofisher). Cells we analyzed using CellProfiler [42] by counting all green staining in the area overlapping with the nuclear DAPI stain, to obtain mean green signal per cell.

### 4.13. Statistical Analysis

All statistical analysis and calculations have been done using software GraphPad Prism 5. Unpaired *t*-tests with 95% confidence interval was performed between different treatment group (* *p* < 0.05, ** *p* < 0.01, *** *p* < 0.001).

## Figures and Tables

**Figure 1 ijms-20-06265-f001:**
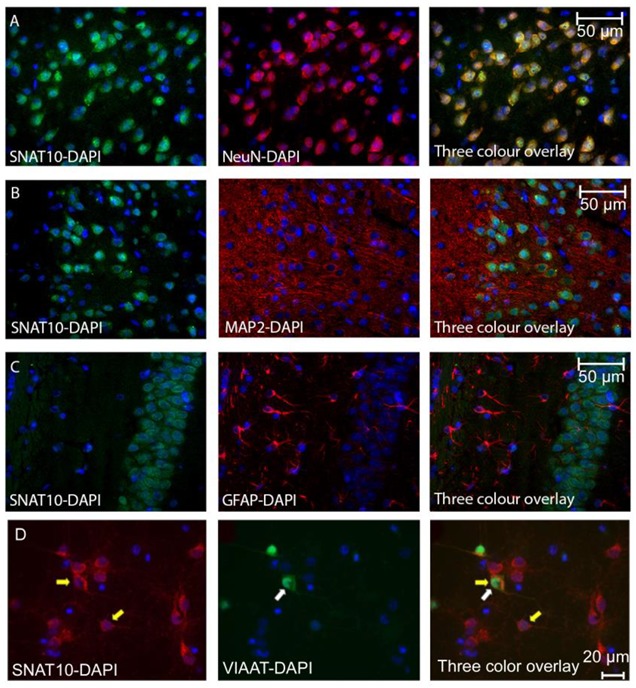
(**A**) Mouse brain sections stained with anti-SNAT10 (green), anti-NeuN (red), and combined images (right most micrograph). Staining of cell bodies with SNAT10 is obvious, with an almost 100% overlap with the anti-NeuN antibody, although anti-SNAT10 staining appears more prominent close to the cell nuclei. (**B**) Mouse brain sections stained with anti-SNAT10 (green), anti-MAP2 (red) and combined images (right most micrograph). Staining of anti-SNAT10 and anti-MAP2 appears in the same cells, but with no *in situ* overlap, suggesting that SNAT10 is not expressed in any neuronal projections. (**C**) Mouse brain sections stained with anti-SNAT10 (green), anti-GFAP (red) and combined images (right most micrograph). No overlap between GFAP and SNAT10 is observed suggesting no expression of SNAT10 in GFAP positive glia cells. (**D**) Fluorescence micrograph of primary cell culture from adult mouse brain where the inhibitory neurons are marked with eGFP (green) and custom-made anti-SNAT10 antibody (Innovagen) with Alexa Fluor 594 (red). The nucleus is stained in blue using DAPI. Left most micrograph illustrates SNAT10 staining in red with two yellow arrows and the nuclei stained with DAPI are in blue. Middle micrograph with white arrow shows eGFP expressing inhibitory neuron in green and the nuclei in this micrograph are in blue. Right most micrograph shows a three color overlay with overlap between green and red staining indicating expression of SNAT10 in the inhibitory neurons with both yellow and white arrows. The yellow arrow indicates a cell marked with only red, revealing expression of SNAT10 in a neural cell other than inhibitory neuron.

**Figure 2 ijms-20-06265-f002:**
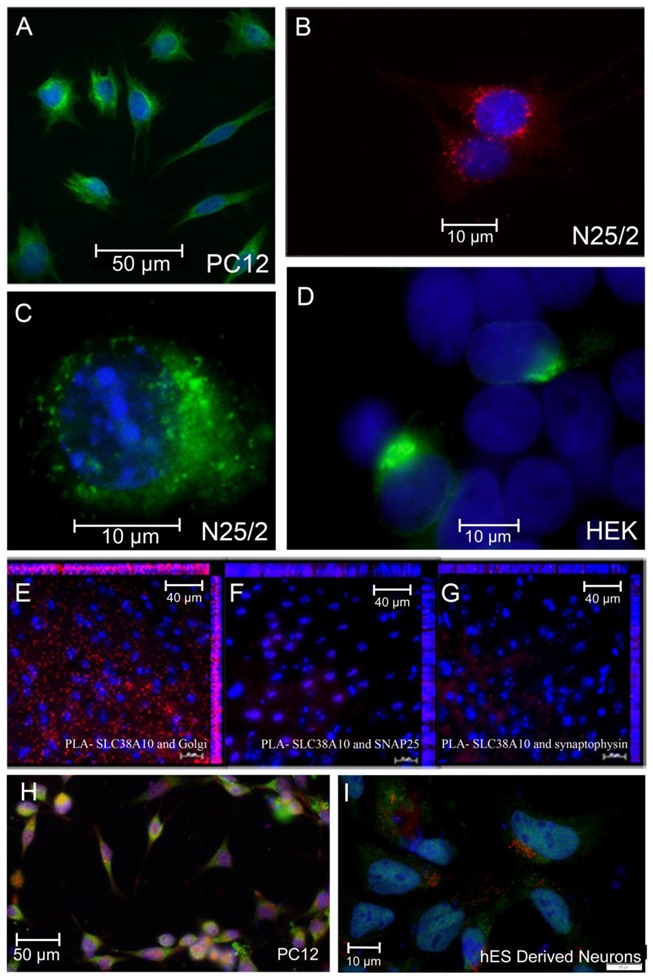
Fluorescence micrograph of PC12, N25/2, and HEK293 cells expressing SLC38A10 illustrating its intracellular localization adjacent to nuclei and proximity ligation assays revealing different degrees of interaction between SNAT10 and Golgi protein, SNAP25 and synaptophysin, respectively. (**A**) PC12 cells stained with custom-made SNAT10 antibody (Innovagen) with Alexa Fluor 488 (green) and nuclei stained with DAPI are shown in blue. (**B**) N25/2 cells stained with custom-made SNAT10 antibody (Innovagen) with Alexa Fluor 594 (red) and nuclei stained with DAPI are shown in blue. Using of two different secondary antibodies (Alexa Fluor 488 and Alexa Fluor 594) in (**A**,**B**) respectively nullifies any effect that the secondary antibodies might have had on the cell lines and therefore shows true localization of SNAT10. (**C**) Fluorescence micrograph of N25/2 cells transfected with SNAT10-eGFP construct showing localization of SNAT10 adjacent to the nuclei that are stained with DAPI and shown in blue. (**D**) Fluorescence micrograph of HEK293 cells transfected with SNAT10-eGFP construct showing localization of SNAT10, adjacent to the nuclei that are stained with DAPI and shown in blue. Both (**C**,**D**) show the expression of SNAT10 at one of the edges of the nuclei. (**E**) Micrograph visualizing proximity ligation assay (PLA) signals in red between SNAT10 and Golgi protein. (**F**) Micrograph visualizing PLA signals in red between SNAT10 and SNAP25. (**G**) Micrograph visualizing PLA signals in red between SNAT10 and synaptophysin. In (**E–G**) nuclei are stained with DAPI and shown in blue. The micrographs (**E–G**) show a comparison between Golgi protein, SNAP25, and synaptophysin in relation to SNAT10. It is obvious that SNAT10 is in proximity to Golgi proteins and not SNAP25 or synaptophysin. (**H**) PC12 cells transfected with a FLAG-tagged SNAT10 construct and stained with anti-FLAG antibody (green) and Golgi protein (red) as a marker for the Golgi apparatus. (**I**) Human embryonic stem cell-derived neurons transfected with a FLAG-tagged SNAT10 transgene and stained with anti-FLAG antibody (green) and Golgi protein (red).

**Figure 3 ijms-20-06265-f003:**
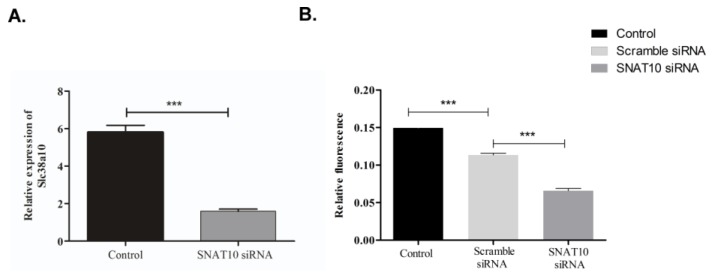
(**A**) siRNA against SNAT10 induces a significant reduction in SNAT10 mRNA expression, measured with qRT-PCR. SNAT10 expression was normalized against β-tubulin expression. Unpaired t-test with 95% confidence interval was performed, (*** *p* < 0.0001) (*n* = number of samples, SNAT10siRNA *n*= 2, control *n* = 3, scramble siRNA *n* = 3). (**B**) After *Slc38a10* knockdown a significant reduction (*p* < 0.001) in protein translation rate, measured using OPP 488 protein translation synthesis on cells treated with OPP, incorporated into nascent peptides was visualized using fluorescent Click-iT chemistry. Unpaired t-test with 95% confidence interval was performed, (*** *p* < 0.0001) (*n* = number of cells counted in analysis, SNAT10 siRNA *n* = 957, control *n* = 2280, scramble siRNA *n* = 2708).

**Table 1 ijms-20-06265-t001:** Details of antibodies used for fluorescent immunocytochemistry and PLA.

Primary Antibodies	Species	Dilution	Company
Anti-SNAT10	Rabbit	1:100	Innovagen
Anti-Synaptophysin	Mouse	1:250	BD Transduction lab
Anti-SNAP25	Mouse	1:500	Millipore
Anti-Golgi 58k	Mouse	1:50	AbCam
Anti-KDEL	Mouse	1:200	Abcam
Anti-GFP	Chicken	1:200	Millipore
Anti-FLAG	Mouse/Rabbit	1:200	Invitrogen
Secondary antibodies	Species	Dilution	Company
Anti-rabbit-594	Donkey	1:800	Invitrogen
Anti-mouse-488	Goat	1:800	Invitrogen

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
