# Peer review of "SLC38A10 (SNAT10) is Located in ER and Golgi Compartments and Has a Role in Regulating Nascent Protein Synthesis"

_ijms, 2019, doi:10.3390/ijms20246265_

Round 1

Reviewer 1 Report

This manuscript describes an interesting study in which the subcellular localisation of the SLC38 family transporter SNAT10 has been investigated in a series of mammalian cell lines, including neuronal culture models. The main conclusions are that (unlike all previously described amino acid transporters) the SNAT10 protein is mainly located within the ER and Golgi (especially the Golgi) and that, from this location, it unexpectedly exerts a significant stimulatory effect on translation and global protein synthesis (which was demonstrated by successful siRNA silencing of SNAT10 mRNA expression within PC12 cells).

In my view this study has been performed in a technically competent way (as shown by the extensive Methods section) and reports potentially important findings. However, in view of the somewhat unexpected observation of a Golgi membrane protein signalling to global protein synthesis, I think that it would be helpful if the authors could refute and/or discuss the following alternative explanations of their findings:

Major points

1)    Many of the cell culture experiments reported in this paper were performed in the presence of Amphotericin B (Section 4.1) which has been widely reported to exert significant effects on membrane structure in mammalian cells (as well as in its intended fungal target cells). Such effects of Amphotericin B may include marked changes in the ultrastructure of the Golgi apparatus (Yamaguchi, H et al Golgi membrane-associated degradation pathway in yeast and mammals. The EMBO Journal (2016) 35: 1991–2007. DOI 10.15252/embj.201593191). Could the reported strong localisation of SNAT10 in the Golgi region (but not in the plasma membrane) be an in vitro artefact arising from blockade of membrane protein trafficking between the Golgi and the plasma membrane by Amphotericin B? Were any of the key experiments in this manuscript performed without this drug? If so it would be helpful if the authors could state this clearly, and use this to refute the artefactual explanation outlined above.

2)    Even if SNAT10 is largely located in the Golgi, it is possible in principle that a small (but functionally very active) pool of the SNAT10 protein occurs elsewhere in the cells (e.g. in the plasma membrane (as for SNAT2) or in association with mTORC1 in lysosomes (as for SNAT9)). I suggest that the authors should discuss and/or refute this as an alternative explanation of their observation of SNAT10 silencing impairing translation in Figure 3. In these siRNA silencing experiments, did the authors perform any parallel measurements of radio-labelled L-glutamine transport across the plasma membrane into intact cells, or measurements of mTORC1 signalling which might have detected such plasma membrane or lysosomal functional effects?

3)    In view of the interesting finding of apparent coupling between SNAT10 expression and translation in Figure 3B, it seems important to confirm that this is genuinely a direct effect of SNAT10 expression. Were parallel measurements of mRNA expression of other SNAT(SLC38) genes performed on the RNA extract that was obtained in Figure 3A, and what was the final concentration of the silencing siRNA that was present in the culture medium? It would be re-assuring to see that the concentration of silencing siRNA that was used here was not exerting off-target effects by silencing SNAT2 or SNAT9.

4)    In the Discussion section of the manuscript the authors raise the interesting possibility that the C-terminal domain of SNAT10 may serve as the sensor component of a transceptor, thus potentially allowing the SNAT10 protein to generate a signal to regulate translation. What is the orientation of the SNAT10 protein in the Golgi membrane? Does the C-terminal domain probe the lumen of the Golgi sacs or the cytosol? If the orientation of SNAT10 in the Golgi membrane is the same as the presumed orientation of SNAT2 when it passes through the Golgi (i.e. with the C-terminus in the Golgi lumen, ultimately yielding a plasma membrane protein with an extracellular C-terminal domain), does this mean that SNAT10 is probing the amino acid pools inside the Golgi sacs? If so, can the authors suggest how such a transceptor regulates translation? For example, is there any evidence in the literature that the size of intra-Golgi amino acid pools can influence translation or global protein synthesis rate?

Minor points

5)    The manuscript contains a large number of typographical errors and errors in the sentence structure (especially in the Abstract and in the main text of the Introduction and Results sections). These need to be corrected.

6)    The title contains the word “co-localized” which implies that SNAT10 is located in association with some other protein which has not been stated. It may be simpler to replace this word with “located”.

7)    In the first sentence of the Introduction it seems to be stated that all 430 SLC transporters are secondary active transporters, rather than (for example) passive transporters. Is this true?

8)    In the legend for Figure 3 the number of biological replicates (n = 2?) should be stated.

9)    In the cell culture section (Section 4.1) the volumes of supplements that were added to the culture medium are stated, but the corresponding volume of basal culture medium is not stated. Presumably this was 500ml.

10)  Section 4.3 is entitled “Primary cell culture” but the exact type of cell that is being grown here  is not stated.

Author Response

Reviewer #1:

1)    Many of the cell culture experiments reported in this paper were performed in the presence of Amphotericin B (Section 4.1) which has been widely reported to exert significant effects on membrane structure in mammalian cells (as well as in its intended fungal target cells). Such effects of Amphotericin B may include marked changes in the ultrastructure of the Golgi apparatus (Yamaguchi, H et al Golgi membrane-associated degradation pathway in yeast and mammals. The EMBO Journal (2016) 35: 1991–2007. DOI 10.15252/embj.201593191). Could the reported strong localization of SNAT10 in the Golgi region (but not in the plasma membrane) is an in vitro artifact arising from blockade of membrane protein trafficking between the Golgi and the plasma membrane by Amphotericin B? Were any of the key experiments in this manuscript performed without this drug? If so, it would be helpful if the authors could state this clearly, and use this to refute the artefactual explanation outlined above.

Thank you for the observation. This part was of materials and methods remained from the previous version of the manuscript, and as a response to previous comments these cell culture experiments have been remade, this time with Amphotericine B. As for the current experiment we did not use Amphotericine B for any of the experiments. We have corrected this in the manuscript.

Even if SNAT10 is largely located in the Golgi, it is possible in principle that a small (but functionally very active) pool of the SNAT10 protein occurs elsewhere in the cells (e.g. In the plasma membrane (as for SNAT2) or in association with mTORC1 in lysosomes (as for SNAT9)). I suggest that the authors should discuss and/or refute this as an alternative explanation of their observation of SNAT10 silencing impairing translation in Figure 3. In these siRNA silencing experiments, did the authors perform any parallel measurements of radio-labelled L-glutamine transport across the plasma membrane into intact cells, or measurements of mTORC1 signalling which might have detected such plasma membrane or lysosomal functional effects?

Its really good point and we have added this to the discussion. Regarding the measurement of radio-labelled L glutamine uptake and motor signalling. We haven't done any such experiment after siRNA knockdown. As we already know SLC38A10 is bidirectional transporter of glutamine, glutamate and aspartic acid (Hellsten et al., 2017), so it is a possibility that we do see an effect on uptake of thise amino acids, providing we have SNAT10 is expressed in the plasma membrane. As previously published by many groups, glutamine is a well known activator of the mTOR pathway(Jewell et al., 2015). We have however not followed this up in the current manuscript, but it would be an interesting future study.

 In view of the interesting finding of apparent coupling between SNAT10 expression and translation in Figure 3B, it seems important to confirm that this is genuinely a direct effect of SNAT10 expression. Were parallel measurements of mRNA expression of other SNAT(SLC38) genes performed on the RNA extract that was obtained in Figure 3A, and what was the final concentration of the silencing siRNA that was present in the culture medium? It would be re-assuring to see that the concentration of silencing siRNA that was used here was not exerting off-target effects by silencing SNAT2 or SNAT9.

We very much appreciate the reviewer’s suggestion. We did not do thiese measurments, and it would have been an interesting thing to do. Howevere, regardless of the results of these experiments, there would have been several possible explainations. If we would have seen a change in regulation of other SNATs, this could be due to off target effects or du to an expressino chagnge in reponse to downregulation of SNAT10. We find the risk of off-target effects on other SNATs limited, the concentration of siRNA was 0.01 pmol/ml (we have added this to the materials and methods) and the degree of conservatino between SNAT10 and the other SNATs are really low, globaly below 10% identity at the amino acid level. It is likely that SNAT10 has a role in regulating metabolism and in that context it will mostel ikely regulate alos expression levels of other SNATs.

In the Discussion section of the manuscript the authors raise the interesting possibility that the C-terminal domain of SNAT10 may serve as the sensor component of a transceptor, thus potentially allowing the SNAT10 protein to generate a signal to regulate translation.

SNAT10 predicted structure has been published by our lab (Hellsten et al.), Prediction of transmembrane helices for SLC38A10 has an intracellular N-terminal and a long C-terminal of 722 amino acids (amino acid 398–1019) on the outside of the membrane. A model of thepredicted transmembrane folding of the SLC38A10 protein sequence with 11 TMHs is presented in below figure. It is a very interesting future study to investigate protein-protein interaction between the C-terminal tail of SNAT10 and possible binding proteins. It would also be interesting to investigate, using truncated forms of SNAT10, if the C-terminus is involved in regulation of intracellular localization.

What is the orientation of the SNAT10 protein in the Golgi membrane? Does the C-terminal domain probe the lumen of the Golgi sacs or the cytosol? If the orientation of SNAT10 in the Golgi membrane is the same as the presumed orientation of SNAT2 when it passes through the Golgi (i.e. with the C-terminus in the Golgi lumen, ultimately yielding a plasma membrane protein with an extracellular C-terminal domain), does this mean that SNAT10 is probing the amino acid pools inside the Golgi sacs? If so, can the authors suggest how such a transceptor regulates translation? For example, is there any evidence in the literature that the size of intra-Golgi amino acid pools can influence translation or global protein synthesis rate?

This is actually very interesting comment and we would really like to further investigate the structure of SNAT10. But currently we are not sure about SNAT10 C-terminal orientation is it towards Golgi membrane, and even if this is known we have shown that SNAT10 is a bidirectional transporter (Hellsten et a.,l 2017) making it hard to predict the outcome of the changes of the amino acids in the Golgi lumen. One possibibity is that SNAT10 is directly interacting with downstream signaling pathways, possibly through its long C-terminal domain. On possibility is that such an interaction would be with members of for example the mTORC1 pathway.

Minor points

5)   The manuscript contains a large number of typographical errors and errors in the sentence structure (especially in the Abstract and in the main text of the Introduction and Results sections). These need to be corrected.

Thank you so much for your comment. We have addressed it during the revision of the manuscript.

6)   The title contains the word “co-localized” which implies that SNAT10 is located in association with some other protein which has not been stated. It may be simpler to replace this word with “located”.

Thank you very much for your valuable suggestion. We have made changes in the title of the revised manuscript.

7)   In the first sentence of the Introduction it seems to be stated that all 430 SLC transporters are secondary active transporters, rather than (for example) passive transporters. Is this true?

Thank you for the comments. Yes ,this is right this are called secodary active transproter (Kandasamy et al., 2018), but formally that is not fully true, the majority of SLCs are secondary active transporters, but there are also passive transporters within this family. We have removed that statement to avoid confusion.

8)   In the legend for Figure 3 the number of biological replicates (n = 2?) should be stated.

We have replaced it with new figure with changes in legend for better understanding. Hopefully you will find it justified.

9)   In the cell culture section (Section 4.1) the volumes of supplements that were added to the culture medium are stated, but the corresponding volume of basal culture medium is not stated. Presumably this was 500ml.

Thank you for mentioning it. Yes it is corresponds to 500 ml Basal culture media.

We have followed Manufacturer (ATCC) recommendations: F-12K Media supplemented with 15 % Horse Serum (HS) + 2.5 %Fetal Bovine Serum (FBS). We have clarified this in the manuscript.

10) Section 4.3 is entitled “Primary cell culture” but the exact type of cell that is being grown here is not stated.

Thank you very much for noting, we have clarified that in the manuscript.

Reviewer 2 Report

I read the paper entitled SLC38A10 is Co-localized in ER and Golgi compartments and has a role in regulating nascent protein synthesis by Tripathi et al.

The manuscript is well written and organized.  Interesting findings.

I only have the following minor comments

Line 28  A sentence should not start with a number, it should read Eleven

Line 71 or abscent (should read absent), consider removing leaving it to read very low

Line 78 absolut should read absolute but it is not necessary, consider removing

line 207  cell culture, please include the passage numbers and from where you obtained the cells

Line 329 10 pmol can be removed, listed twice

Author Response

Reviewer #2:

1. Line 28  A sentence should not start with a number, it should read Eleven.

Thank you for the comment we have made changes accordingly.

2. Line 71 or abscent (should read absent), consider removing leaving it to read very low

Thank you for the comment. We have tried to reconstruct more clear with the new statement.

3. Line 78 absolut should read absolute but it is not necessary, consider removing 

Thank you for pointing out we have made changes accordingly.

4. line 207  cell culture, please include the passage numbers and from where you obtained the cells.

Thank you for the comment we have made changes accordingly.

5. Line 329 10 pmol can be removed, listed twice

Sorry for the repetition and we have edited it as per suggested.

Reviewer 3 Report

Tripathi and colleagues (ijms-650803) describe for the first time the expression of the tenth member of the SLC38 transporter family in the mouse brain, and state its role in amino acid supply and regulation of protein synthesis in neuronal cells. The transporter protein is mainly expressed in the Golgi apparatus where it might act as transceptor to sense signals for modulation of protein synthesis and protein modification. The scientific message of the manuscript is new and will be of interest for neurologist, physiologists and development biologists, however, the text needs severe revision. The manuscript leaves many points open (see following comments) and therefore cannot be accepted for publication.

Comments

Figure 2: Please delete the 10 µm scale bar in Fig. 2A.

Figure 3: Include the number of independent replicates in the figure legend. Which statistical test was used? In Fig. 3B the labeling of the x-axis should be at the same level.

Discussion: The first paragraph of the discussion differs significantly from the following text. Especially lines 173 to 176 are fragmented and must be revised. I don’t understand the sentence in line 200. What is meant with “…Knocking down the expression of Slc38a10 in PC12 cells resulted in a protein synthesis process,…”?

Materials and Methods:

Point 4.1: As most chemicals/media are from Gibco, it can be stated in one sentence like “All chemicals/media were from Gibco, Stockholm, Sweden, except another company name is stated”.

Point 4.2: The concentration of the plasmid in DMEM should be presented in µg/ml.

Point 4.4: Subheadings should be avoided (see also point 4.6).

Point 4.8: The sentence in lines 329/330 is hard to understand. How was the negative control treated: with scrambled siRNA or just with Lipofectamine?

Point 4.10: Which statistical tests were used? Please include a precise description of the statistical analysis.

All antibodies should start with “anti-“. The antibody names in the text and in table 1 should be equal.

The text is written in an inattentive way and contains many oversights, which makes it extremely difficult to concentrate on the scientific message. Please control the following words: transceptor, fluor, fluorescence, transporter, poly-L-lysine, …, paraformaldehyde, formaldehyde. Choose a consistent wording and style throughout the text.

Author Response

Reviewer 3:

1. Figure 2: Please delete the 10 µm scale bar in Fig. 2A.

Thank you so much for your minute observation and valuable comments. It really helped and we have made correction to figure.

2. Figure 3: Include the number of independent replicates in the figure legend. Which statistical test was used? In Fig. 3B the labeling of the x-axis should be at the same level.

As suggested details has been added into legend. Unpaired t-tests with 95% confidence interval was performed between different treatment group *P < 0.05,**P < 0.01, ***P < 0.001).)

3. Discussion: The first paragraph of the discussion differs significantly from the following text. Especially lines 173 to 176 are fragmented and must be revised. I don’t understand the sentence in line 200. What is meant with “…Knocking down the expression of Slc38a10 in PC12 cells resulted in a protein synthesis process,…”?

Thank you for we have tried to reconstruct statement for better undersatanding.

Materials and Methods:

4. Point 4.1: As most chemicals/media are from Gibco, it can be stated in one sentence like “All chemicals/media were from Gibco, Stockholm, Sweden, except another company name is stated”.

Thank you for the suggestion, this has been corrected in the revised MS.

5. Point 4.2: The concentration of the plasmid in DMEM should be presented in µg/ml.

Thank you so much for your comment. We have addressed it during the revision of the manuscript.

6. Point 4.4: Subheadings should be avoided (see also point 4.6).

We have made changes as suggested.

7. Point 4.8: The sentence in lines 329/330 is hard to understand. How was the negative control treated: with scramble d siRNA or just with Lipofectamine?

Negative controal is treated with scramble where as mock control only transfection reagent was added. We have rewritten this section to make it more clear.

8. Point 4.10: Which statistical tests were used? Please include a precise description of the statistical analysis.

Response: Thank you for pointing out .Unpaired t-tests with 95% confidence interval was performed between different treatment group and details has been included in method part. We have also added this to the legend for figure 3.

9 .All antibodies should start with “anti-“. The antibody names in the text and in table 1 should be equal.

The text is written in an inattentive way and contains many oversights, which makes it extremely difficult to concentrate on the scientific message. Please control the following words: transceptor, fluor, fluorescence, transporter, poly-L-lysine, …, paraformaldehyde, formaldehyde. Choose a consistent wording and style throughout the text.

All of these comments and suggestions were taken into consideration and were corrected in the revised MS.

Round 2

Reviewer 1 Report

The authors have adequately addressed the points that I raised in my original report.

Reviewer 3 Report

The authors have responded to all points raised and have significantly increased the text of the manuscript. After additional minor editing of the present text, the manuscript can be accepted for publication.